Age and lunar cycle predict harbor porpoise bycatch in the south-western Baltic Sea

Brennecke Dennis 1 2 3
Wahlberg Magnus 2
Gilles Anita 1
Siebert Ursula ursula.siebert@tiho-hannover.de 1
1 Institute for Terrestrial and Aquatic Wildlife Research, University of Veterinary Medicine Hannover, Foundation , Büsum , Schleswig-Holstein , Germany
2 University of Southern Denmark, Department of Biology, Marine Biological Research Centre , Kerteminde , Denmark
3 Leibniz Institute for Science and Mathematics Education , Kiel , Schleswig-Holstein , Germany
Garant Dany
Electronic publication date: 2021 Oct 27
Publication date: 2021
Volume: 9
Electronic Location ID: e12284
Received 2021 Mar 10; Accepted 2021 Sep 20
Copyright: ©2021 Brennecke et al.
Copyright year: 2021
Copyright holder: Brennecke et al.
License: This is an open access article distributed under the terms of the Creative Commons Attribution License, which permits unrestricted use, distribution, reproduction and adaptation in any medium and for any purpose provided that it is properly attributed. For attribution, the original author(s), title, publication source (PeerJ) and either DOI or URL of the article must be cited.
License URL: https://creativecommons.org/licenses/by/4.0/

Keywords: Harbor porpoise, Phocoena phocoena, Fisheries, Bycatch, Baltic Sea, Abiotic factors, Lunar cycle, Marine mammals, Cetacean

Funding: The Ministry of Energy, Agriculture, Enviroment, Nature and Digitalisation of Schleswig-Holstein The collection of animals was funded by the Ministry of Energy, Agriculture, Enviroment, Nature and Digitalisation of Schleswig-Holstein. There was no additional external funding received for this study. The Ministry of Energy, Agriculture, Enviroment, Nature and Digitalisation of Schleswig-Holstein funded animal collection activities.

==============================
The harbor porpoise, Phocoena phocoena, is the only cetacean regularly occurring in the Baltic Sea. During the last decades, several anthropogenic activities have affected porpoises in the Baltic region. Most notably is bycatch in static fishing gear, such as gill nets, which is the main human-induced cause of death in odontocetes. There is still considerable uncertainty about which factors influence the amount of bycatch. In the present study, we reviewed bycatch data collected from 1987 to 2016 from the south-western Baltic Sea. There was a significant difference in bycatch due to seasonality and region, and there was a higher bycatch rate in juveniles than in adults. The only abiotic factor associated with bycatch was the lunar cycle, with more animals bycaught during a full moon. These results improve our understanding of which biotic and abiotic factors are associated with bycatch of Baltic harbor porpoises, which can be used to strengthen conservation endeavors such as managing fishing efforts.

Introduction

The Baltic Sea is severely altered by anthropogenic activities and, therefore, one of the most intensely studied coastal areas worldwide (Korpinen et al., 2012; Reusch et al., 2018). Intensive fisheries, shipping, military activities, and pollution are collectively notorious threats to Baltic Sea fauna (HELCOM, 2018). Incidental catches during fishing operations, i.e., bycatch, are considered the greatest global threat to cetaceans (Read, Drinker & Northridge, 2006; Reeves, McClellan & Werner, 2013); especially to small cetaceans, like the harbor porpoise, Phocoena phocoena (Brownell et al., 2019). It is therefore highly likely that bycatch has a substantial impact on the population size of Baltic harbor porpoises (van Beest et al., 2017).

There have been considerable conservation efforts regarding this species due to its vulnerability to human disturbance (Siebert et al., 2006; Carlén et al., 2018; Carlén, Nunny & Simmonds, 2021). For all three populations in the Baltic region, including the North Sea population that occurs from northern Kattegat to the North Sea, assessed bycatch rates are likely above 1.7% of the population size, which by ASCOBANS has been defined as the maximum sustainable level for anthropogenic mortality (Vinther, 1999; ASCOBANS, 2000; Berggren et al., 2004; Vinther & Larsen, 2004; Bjørge, Skern-Mauritzen & Rossmann, 2013; Moan et al., 2020). Even though some of these bycatch estimates are old and associated with large errors, they call for great concern regarding the impact of bycatch of both small and large populations of harbor porpoises (Rogan, Read & Berggren, 2021). In addition, it has been estimated that the critically endangered Baltic Proper population can only sustain a maximum of 0.7 animals per year killed by human activities (North Atlantic Marine Mammal Commission and the Norwegian Institute of Marine Research, 2019). In case of such small and threatened populations any bycatch of animals may therefore be disastrous for the population. Efficient solutions to the porpoise bycatch problem involve changing fishing practices, including time-area fishing closures (van Beest et al., 2017), use of alternative fishing gear (Königson et al., 2015), or improving the acoustic detectability of gillnets (Kratzer et al., 2020). To make mitigation measures as efficient as possible, it is important to know the environmental context in the area where bycatch is occurring. For example, the soundscape experienced by the animals may impact the risk of bycatch. Animals may be distracted by increased levels of sound, or such sounds may mask the porpoises’ abilities to detect the gill nets by echolocation. Negative effects of underwater noise disrupting the behavior of harbor porpoises have been documented (Tougaard, Henriksen & Miller, 2009; Dähne et al., 2013; Dyndo et al., 2015; Wisniewska et al., 2018). It is conceivable that also natural sounds can disrupt harbor porpoise behavior. For example, noise generated by extreme weather events, such as heavy rain can increase noise levels by up to 50 dB above the background noise level across a broad range of frequencies (Nystuen, 1986). In addition, wind may also create distracting sounds (Pensieri et al., 2015). Furthermore, there are many environmental drivers for fish activity such as anti-predator and food search behavior on a fine scale, and diel and seasonal rhythms on a larger scale (Løkkeborg & Fernö, 1999). Many species of fish have activity patterns steered by moon phases, and therefore it is conceivable that also porpoise behavior could be affected by changes in lunar illumination, which may therefore affect the risk of bycatch (Takemura et al., 2004; Horký et al., 2006; Celik & Celik, 2011). Lunar influence on cetaceans has been observed in indo-pacific humpback dolphins (Wang et al., 2015) and also in the dive behavior and spatial area used by pilot whales (Owen et al., 2019). Thus, perhaps lunar cycles may also influence porpoises.

To date, it is unknown which abiotic, biotic and/or anthropogenic factors might correlate with the risk of a porpoise being bycaught. In the present study, we analyzed a long-term dataset of 30-year bycatch data from the south-western Baltic Sea. Besides seasonal and spatial trends, we investigated whether age and sex play an important role in the likelihood of animals being bycaught. In addition, we examined whether harbor porpoise abundance and gillnet fishing activities affect bycatch. The role of environmental conditions, such as wind, precipitation and lunar cycle on bycatch events was further evaluated.

Materials & Methods

Data collection

From 1987 to 2016, 140 bycaught harbor porpoises, collected by the stranding network of Schleswig-Holstein along the German Baltic Sea coast (Siebert et al., 2006), were handed over by fishermen and brought to the Institute for Terrestrial and Aquatic Wildlife Research (ITAW), University of Veterinary Medicine Hannover, Foundation, Germany for necropsy (Siebert et al., 2001; Siebert et al., 2020; Wünschmann et al., 2001). Fishermen are obliged to report bycatch of cetaceans by law. Data on location, date of bycatch, age, body length, weight and sex of the bycaught animals were collected. Age was preferably determined using teeth by counting the annual growth layers, or (if dental age determination was not possible) by measuring the body length of the animals (Myrick et al., 1983; Kremer, 1987; Lockyer, 1995; Siebert et al., 2001). Animals were categorized in age groups based on Siebert (2001; 2020): Neonates and calves: ≤0.5 years old, juveniles/subadults: 0.5–4 years old and adults: >4 years old. Four animals were excluded from the data set for the seasonal distribution analysis due to inconclusive age determination. Only bycatch events notified directly by fishermen and confirmed by necropsy were used. While reporting a bycatch event, a few fishermen also reported the used net type (single wall and three wall gill nets), target fish (mainly cod and flatfish) and water depth (10–25 m).

In order to put reported bycatch events in the context of harbor porpoise abundance estimates in the region, data from aerial surveys conducted regularly in the area following line-transect distance sampling methodology, were used. Hereby, only surveys conducted in the stratum Kiel Bight from 2002–2015 were considered (see stratum “E” in Scheidat et al., 2008). To evaluate the possible seasonal change between bycatch and fishing effort, we included data from an analysis of the fisheries in the south-western Baltic Sea between 2010 and 2012 using monthly averages of fishing effort (Dorrien et al., 2013). Only fishing boats larger than 8 m using gillnets were considered, since fishermen with these boats are obliged to report fishing activities effort, that can later be used to calculate the fishing effort, in a logbook.

In order to determine, if weather conditions could have been associated with bycatch events, we analyzed meteorological data provided by the German National Meteorological Service (Deutscher Wetterdienst, DWD). During the time period from 1987 to 2016 we obtained information on daily precipitation (mm), daily maximum wind speed data up to 15 m/s and mean daily cloud coverage (oktas) from nine different weather stations (Kiel lighthouse, Fehmarn, Glücksburg, Schönhagen, Kiel-Holtenau, Travemünde, Flensburg, Hohwacht, Lübeck Blankensee) closest in distance to each bycatch event. Weather data was not available for each bycatch event.

We also analyzed if the lunar cycle correlated with bycaught animals. Data on the lunar cycle were obtained from the US Naval Observatory (USNO, https://www.usno.navy.mil/USNO) at 54°30′N, E10°20′E (this location is within 40 nm from any of the bycatch events studied here), one hour ahead of Greenwich Mean Time (GMT) which corresponds to German time without daylight saving. For each bycatch event, the percentage of illumination of the moon’s visible disk was categorized into eight phases as follows: (i) new moon; (ii) waxing crescent; (iii) first quarter; (iv) waxing gibbous; (v) full moon; (vi) waning gibbous; (vii) last quarter; and (viii) waning crescent.

Statistical analyses

We conducted regressions using polynomials of different orders for the bycatch time series to assess whether bycatch had changed over the 30-year period and whether bycatch changed with porpoise abundance between 2002 and 2015. We used partial F-tests with the ANOVA command in R to determine which polynomial model provided the best fit.

We checked for a statistical significance difference using Chi-squared tests between the bycatch events in (a) different years, (b) different months and (c) different regions, and (d) during different lunar phases in which bycatch occurred. Five animals were excluded from the spatial analysis due to unknown bycatch location. For all Chi-squared tests we calculated the expected frequencies by dividing the total sample size by the number of groups. In order to test for significant differences between wind speed and precipitation during days with and without bycatch, we used ANOVA. Statistical analyses were performed using R software, version 3.5.2 (R Core Team, 2018). Data visualization was conducted by “ggplot2” package, version 3.2.1, in R software (Wickham, 2016).

Results

The highest number of bycaught animals was observed in the first decade of the time series (1987–1996), with an average of 7.4 individuals per year, reaching a maximum of 22 animals in 1991 (Fig. 1, X2 = 110, N = 26, 140), p < 0.0001). From 2015 onwards, bycatch increased to more than eight animals per year. Polynomial regression analysis revealed a significant relationship between the number of bycaught animals and year (R2 = 0.5, F5,21 = 4.2, p < 0.01). When seasonal differences were investigated by binning the numbers of bycaught porpoises by different months, 69.3% of all porpoises were bycaught from July to November, with the highest number observed in August with 29 animals (X2 = 61, N = 11, 140, p < 0.0001). Juveniles were bycaught more frequently (73%) followed by adults (20%) and neonates (7%). Most juveniles (71%) were bycaught between July and November. Neonates were only bycaught between July and October (Fig. 2). When comparing monthly binned data for female and male bycaught animals, there was no significant difference in the amount of bycaught animals (i.e., males: 47% and females: 53%, X2 = 0.6, N = 1, 139, p = 0.45), except for a 80% female bycatch in May.

Figure 1 Bycatches of harbor porpoises from 1987–2016.

Annual variation in number of harbor porpoise bycatches off the Baltic Sea coast of Schleswig-Holstein from 1987–2016, (n = 140), dashed line: polynomial regression equation: y = 6E − 05x5 − 0..6374x4 + 2552.6x3 − 5E + 06x2 + 5E + 09x − 2E + 12, Multiple R2: 0.51, p < 0.001.

Figure 2 Seasonal distribution of harbor porpoise bycatches.

Seasonal distribution of age groups and sexes of harbor porpoise bycatches off the German Baltic Sea coast of Schleswig-Holstein in the period from 1987–016 (bar chart, n = 136), seasonal distribution of fishing effort in days at sea (solid line, monthly averages of the years 2010–2012; based on Dorrien et al., 2013) and bycatch rate (dashed line, bycatch per fishing effort). Four animals were excluded from the data set due to inconclusive age determination.

The majority of bycatch occurred in Eckernförde Bight and Schlei (Fig. 3 and Table 1; 21.5% and 20% of total bycatch, respectively, X2 = 34, N = 7, 135, p < 0.0001). There was no clear relationship between abundance of porpoises and number of bycaught animals per season between 2002 and 2015 (R2 = 0.24, F2,16 = 2.632, p = 0.10). While gillnet fishing effort increased slightly in the fall from September to November, bycatch was higher from July to November. In August, the bycatch rate, i.e., bycatch per fishing effort, was four porpoises per 100 days at sea (Fig. 2).

Figure 3 Spatial distribution of harbor porpoise bycatches and fishing effort.

Bycatches of harbor porpoises summed per grid cell (10 × 10 km) from 1987–2016. (coordinate system ETRS89LAEA, Pseudo Mercator WGS 84, EPSG 3857). Gray dots indicate harbor porpoise bycatch locations. Fishing boats indicate the annual fishing effort in respective ICES squares (annual average of the years 2010–2012; based on Dorrien et al., 2013).

Table 1 Bycatch of harbor porpoise (Phocoena phocoena) per region.

Bycatch of harbor porpoise (Phocoena phocoena) per region of the south-western Baltic Sea (Schleswig-Holstein) from 1987–2016 (n = 135). Five animals were excluded from this data set due to unknown bycatch location. The regions are ordered from west to east along the coast (see also Fig. 3).

Region	Bycaught porpoises	%	
Flensburg Fjord	22	16.3	
Schlei	27	20.0	
Eckernförde Bight	29	21.5	
Kiel Fjord	13	9.6	
Hohwacht Bight	8	5.9	
Heiligenhafen	19	14.1	
Fehmarn	14	10.4	
Lübeck Bight	3	2.2	
Total	135	100	

There were no significant differences between the maximum wind speed of days with and without bycatch (ANOVA, F1,52787 = 0.003, p = 0.96), and between the daily precipitation during bycatch events and days without any bycatch (ANOVA, F1,68433 = 0.934, p = 0.33; Fig. 4). The majority of bycatch (28.6%) occurred during the lunar phase 5 (full moon; Fig. 5, X2 = 49, N = 7, 140, p < 0.0001). The cloud cover during these full moon bycatch events was in 15% of the cases ≥6 oktas (mean daily cloud coverage). There was more bycatch during moonlight (with lunar illumination >50%) nights (65%) than during dark nights (35%).

Figure 4 The role of environmental conditions on bycatch events.

(A) Maximum daily wind speed measured in m/s at all nine weather stations during bycatch events (n = 75) and during days with no bycatch (n = 52.714). (B) Precipitation in mm measured at all nine weather stations during bycatch events (n = 115) and during days with no bycatch (n = 68.435), y axis is in a Log10 scale.

Figure 5 Number of bycaught harbor porpoises during different lunar phases.

Lunar phase during bycatch events (n = 140) at 54°30′N, 10°20′E, one hour ahead of Greenwich Mean Time (GMT); 1: new moon, 2: waxing crescent, 3: first quarter, 4: waxing gibbous, 5: full moon, 6: waning gibbous, 7: last quarter, 8: waning crescent.

Discussion

Every year, numerous harbor porpoises end up as bycatch in fishing operations. Even though bycatch has been reported for decades, there is still considerable uncertainty regarding possible factors that might be correlated with bycatch. From our data it is clear that bycatch is particularly a major threat to juvenile harbor porpoises in the south-western Baltic Sea, and that the majority of harbor porpoises were bycaught during a full moon and between July and November. Even though the total number of bycaught harbor porpoises remains unknown, bycatch may seriously affect the population size of porpoises in these waters, since the majority of bycaught animals were not sexual mature. The highest number of bycaught animals in our study was observed in the first decade of data collection (i.e., between 1987–1996), when bycatch was more than double the one during the following decades. In the Gulf of Maine, a similar decline in harbor porpoise bycatch was attributed to several mitigation measures, including time-area closures and the use of acoustic alarms, as well as to a declined gill-net fishing effort (Read, Drinker & Northridge, 2006). It is possible that a reduction in bycatch in European waters could have been caused by similar mitigation measures and a similar decline in the gill net fishery. However, it is also possible that regulations of bycatch as well as fishing catch limits in all EU waters initiated in the 1990s and during the last decade in the Baltic Sea to improve the poor state of Baltic cod stocks, have influenced the willingness of fishermen to register bycaught animals. Considering that our data are of opportunistic nature using reported bycatch delivered by fishermen, the reported bycatch decrease may be related to fishermen reporting fewer animals.

To avoid such possible conflicts of interest in the future, a voluntary agreement between fishermen and the Schleswig-Holstein Ministry of Energy, Agriculture, the Environment, Nature and Digitalization was established in 2014 which consists of an anonymous “pick-up service” for bycaught harbor porpoises (OIC, 2015). With 8 animals in 2015 and 10 animals in 2016, this measure seems to make a difference for the willingness of reporting by fishermen. The increase in the number of bycaught animals since 2015 is likely caused by this new agreement rather than a change in the number of animals being caught in fishing gear.

Our study reveals a high bycatch rate of individuals younger than four years old, which is consistent with previous findings (e.g., Berggren, 1994; Kinze, 1994; Kock & Benke, 1996a; Kock & Benke, 1996b; Tregenza et al., 1997). It is likely that the emphasis of young porpoises being bycaught is not only explained by their disability to perceive nets as a threat but also to them being more curios than adults. It is still unknown at which age young animals separate from their mothers, and whether juvenile animals move around alone, or in mixed age groups. It has been reported that during their first 10 months of life harbor porpoise calves spend more than 50% of the time in close proximity with their mothers (Palacino González, 2019).

Considering the lack of experience of juveniles raises the question if there is any learning effect involved in net interactions, meaning that older porpoises are better in avoiding being caught in a net. This would imply that many older animals may have experienced disentanglement themselves from nets when younger and then learned from this dangerous experience, or also if they witnessed others being bycaught and thereby learned to perceive the net as a danger. During necropsies at ITAW over the last 30 years, only a few animals showed scars indicating a previous interaction with fishing gear. In general, assessment of interactions with nets among stranded animals is a difficult task and needs trained veterinary pathologists. Not all bycaught animals show net marks, and for animals in a state of advancing decomposition net marks are difficult to identify (IJsseldijk et al., 2020; Siebert et al., 2020).

The predominance of young porpoises among bycaught animals could also be caused by dietary differences between age groups (Perrin, Donovan & Barlow, 1994; Kastelein, Au & de Haan, 2000). Adult porpoises feed on larger fish, such as flatfish and gadoids, while younger ones prefer smaller sized prey like gobies and sprat (Benke et al., 1998; Wisniewska et al., 2016). Also, the age structure of stranded porpoises is highly biased towards young animals (Siebert et al., 2020). For natural populations, a similar bias towards neonates and juveniles was corroborated by the killing of 41 harbor porpoises, predominately juveniles, through detonations of naval mines in August 2019 (Siebert et al., 2020). Even though harbor porpoises can reach an age of 20–25 years, the average longevity for females in the German Baltic Sea is less than 4 years, i.e., before or shortly after reaching maturity (Kesselring et al., 2018). As there is a predominance of young porpoises in the bycatch as well as in the population, it may also be that the actual bycatch risk for each age class is rather constant.

According to our study, there was no difference in the vulnerability to bycatch between males and females. The high incidence of bycaught adult females in May could be related to the calving season. According to Lockyer (2003) and Hasselmeier (2004), the annual life cycle of harbor porpoises is divided into mating (from July to September), calving (from May to August) and nursing (from September to April). Calving and nursing occur mainly in shallow areas and, thus, females restrict their foraging to those areas and will not dive very deep or search for food in large areas (Koschinski, 2001; Santos & Pierce, 2003; Wisniewska et al., 2016). Many gillnets are found in coastal waters, a preferred habitat for female-calf pairs (Funk et al., 2020). Consequently, the risk of being bycaught in a gillnet is much higher for adult females than males during the first months after birth (Herr, 2009).

Although bycatch occurs year-round, its monthly distribution in this specific area suggests a strong seasonality, with the highest occurrence between July and November, as also found by Kock & Benke (1996a), Kock & Benke (1996b) and Vinther (1999). The most probable reason for the high late summer and fall bycatch is the higher abundance of porpoises in the studied area during the summer months in the waters used for gillnet fisheries. Most sightings of porpoises in the German Baltic Sea are during summer months, however, this is also the season with the highest visual survey effort in general (Gilles, Viquerat & Siebert, 2014; Gilles, Peschko & Siebert, 2011; Gilles & Siebert, 2009; Viquerat et al., 2015; Scheidat et al., 2008).

Regarding the spatial distribution of bycatch events and fishing effort, the majority of bycatch occurred in Eckernförde Bight and Schlei, although the highest fishing effort was in ICES square 37G1, where there was hardly any bycatch. The south-western Baltic Sea is also a habitat for the critically endangered Baltic Proper population during winter months (Benke et al., 2014). Since none of the bycaught animals have been tested genetically, we are not able to determine to which population the bycaught animals belong to. However, even one bycaught harbor porpoise from the Baltic Proper population could have a detrimental impact on this population and a zero anthropogenic mortality limit should be aimed for, also justified by the precautionary principle.

Behavioral changes during summer and fall may affect the risk of porpoises being bycaught in a gillnet. Increased bycatch during increased fishing effort from July and December matched our expectations. In contrast, Herr (2009) and Sonntag et al. (2012) noted a higher number of gillnets in the first six months of the year in the same area, and a lower number in July and August within their observation periods ranging from 2002 to 2008. From their observations, we would expect to have more bycaught animals during the first six months of the year, which is not corroborated by our data. In fact, we found more than 2.5 times lower bycatch during winter and spring and a higher number in the summer and fall. Nevertheless, given that our findings are based on a limited number of observations of fishing activities, the results from such analyses should be treated with considerable caution.

Harbor porpoises can detect gill nets at a range of at least 5–10 m using echolocation (Kastelein et al., 1999); perhaps at even longer ranges (Nielsen et al., 2012). It is not well understood why bycatch occurs even though the nets should be detectable to the animals at longer distances than when they risk entanglement. Foraging, socialization (e.g., distraction by conspecifics) or weather conditions might influence the probability of porpoises to be bycaught. In addition, the Baltic Sea is an environment with high and increasing amounts of anthropogenic activities such as shipping, constructions, military, recreation, and ammunition deposits (Korpinen et al., 2012; Reusch et al., 2018). Noise disturbances can reduce echolocation of porpoises and cause fleeing behaviors (Dyndo et al., 2015; Wisniewska et al., 2018) which may result in nets not being detected and animals becoming entangled. It is unclear if natural sounds may be correlated to bycatch events by creating distractions (e.g., making the nets less audible to the animals by decreasing the detection distance, or by scaring animals so they accidentally swim into the nets). In our study, neither precipitation nor wind speed had any impact on bycatch. Both precipitation and wind speed increase the ambient noise level and could in theory cause the masking of returning echoes from fishing gear, making it difficult for the porpoise to detect them. However, the very high frequency content of the echoes probably prevents wind and rain from having any effect on the ambient noise at these frequencies.

In addition to weather conditions, we investigated the correlation of moon phases with bycatch. The majority of porpoises were bycaught during a full moon and moonlight night. The tidal changes caused by the different phases of the moon are extremely small in the Baltic Sea. Therefore, the effect of the moon on porpoise bycatch seems driven by variations in light conditions rather than by tidal differences. The lunar cycle also affects fish activity (Takemura et al., 2004; Horký et al., 2006; Celik & Celik, 2011). All this together, it is conceivable that porpoises are hunting more actively during the full moon and, consequently, the risk of swimming into a fishing net is higher. In addition, fishing activities depend on weather and perhaps also lunar conditions. If fishing effort increases during the full moon, this could also explain the high bycatch. Clear lunar patterns were observed in the echolocation activity of Indo-Pacific humpback dolphins (Sousa chinensis) and of common dolphins (Delphinus delphis) with higher echolocations detections during the new moon (Wang et al., 2015; Simonis et al., 2017). While no studies have analysed the influence of the lunar cycle on wild harbor porpoises, long-term static acoustic monitoring conducted in the Baltic Sea revealed a 29-day rhythm (i.e., synodic period; M Dähne, 2019, pers. comm). The fact that the clicking activity of captive porpoises is not dependent on lunar cycle (Osiecka, Jones & Wahlberg, 2020) may indicate that this pattern is caused by porpoises adjusting to prey behavior rather than their clicking behavior being directly affected by the lunar cycle. While the reasons for such activity cycles remains unknown, more studies focusing on understanding the influence of meteorological factors on bycatch are needed.

Conclusions

The harbor porpoise is the only cetacean species in the Baltic Sea, and, as a top predator, an important key species and indicator of the health and resilience of the marine ecosystem. The long-term data presented here gives important clues on how environmental parameters in time and space as well as by age- and sex-related factors affects bycatch, a major mortality cause in harbor porpoises. In Germany, to date, there is no established bycatch observation program on small vessels (i.e., <8 m) or for recreational (part-time) fisheries. We may envision that the inclusion of such vessels would make the threat from bycatch even more worrying. In addition, mitigation measures should be updated by including critical time periods (summer months) and areas with high number of bycatch events. To reduce bycatch, fishing efforts could be reduced seasonally between e.g., July and November, especially in areas with high bycatch of juveniles and neonates, such as Eckernförde Bight and Schlei. A more detailed knowledge of the causes of bycatch could result in a more cost-effective regulation of fishing activities. Moreover, considering the potential use of the south-western area in winter by harbor porpoises of the Baltic Proper population, zero bycatch should be allowed. Therefore, future studies on the current status of the Baltic Proper porpoise distribution range (e.g., with genetic studies) are recommended to be able to design management strategies as effective as possible.

Supplemental Information

Supplemental Information 1 Raw data

Click here for additional data file.

The authors wish to thank all dedicated members of the marine mammal stranding response network of Schleswig-Holstein, Germany. Thanks also go to all ITAW colleagues for their support during necropsies and sample collection.

Additional Information and Declarations

Competing Interests

Author Contributions

Data Availability

The authors declare there are no competing interests.

Dennis Brennecke performed the experiments, analyzed the data, prepared figures and/or tables, authored or reviewed drafts of the paper, and approved the final draft.

Magnus Wahlberg and Ursula Siebert conceived and designed the experiments, performed the experiments, authored or reviewed drafts of the paper, and approved the final draft.

Anita Gilles performed the experiments, authored or reviewed drafts of the paper, and approved the final draft.

The following information was supplied regarding data availability:

The raw data are available in the Supplemental File.

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
