# Peer review of "Age and lunar cycle predict harbor porpoise bycatch in the south-western Baltic Sea"

_PeerJ, doi:10.7717/peerj.12284_

## Round 0.1 · original submission · Major Revisions

We have received two reviews for your manuscript. Both reviewers found the study interesting but also raised a number of important issues that deserve major revisions.

Most importantly, both reviewers identified several issues with the current analytical approach that will warrant a more detailed description and an in-depth revision of statistical analyses. For instance, reviewer 1 suggested using a generalized additive model to better assess bycatch rate. Also, a more comprehensive report of data being used is necessary. All other comments provided by both reviewers need to be integrated into the next version.

Reviewer 1 ·

Basic reporting

1. The authors clearly defined their question and motivation.
2. There are some places where the English could be improved, like
line 44: bycatch is not the "main" cause of death of marine mammals, although it may be the main human-induced cause of death.
line 54: "severely affected" needs to be changed.
many places: the term “influencing bycatch” is not totally correct. Consider using terms like “correlated” or “associated”. Although the factor may truly drive (influence) more bycatch, you are actually looking for statistically significant correlations in the data. Do you see the fine difference in terms.
3. The figures look pretty.
4. The discussion section rambles on a bit and is too long (in my opinion). The results section was 24 lines long and the discussion was over 130 lines long.

Experimental design

This manuscript is interested in identifying biotic and abiotic factors that correlate to harbor porpoise bycatch levels, which could be used to inform management actions to reduce bycatch. This is a worthwhile goal, especially for the small Baltic stock.

Unfortunately, the authors need to do more statistical work to actually address this question and thus provide useful information for their managers. Some specific additional analyses and questions about what was done are below.
1) If there were no recorded bycatch from fishermen during 2005, 2010 and 2014 and there was fishing in these years, then these years should be removed from the analyses because it is highly unlikely there were truly no bycatch. As the authors mentioned, the fishers may not have been cooperative those years.
2) Lines 149-150: What was the purpose of developing a polynomial regression line of the time series of annual numbers of bycatch? The real question is why is there interannual variability? Was it because of some biotic or abiotic factor that managers could use to develop a management action?
3) The authors should consider developing a model of the bycatch rate (number of bycatches /100 fishing days at sea) say by month-year (and 10x10 km2 grid cell maybe) as it relates to year, month, age group (perhaps <4 years versus >4 years old), area, sex, wind speed, precipitation, and lunar phase. A generalized additive model would provide the most flexibility. You could look at each variable individually to get a better feel of the relationships. Then you could build a multi-variable model to see which combinations of variables are needed to explain the bycatch patterns. Since many of the factors are related to each other you should check for aliasing.
4) Figure 3 is great, now you need a complimentary map of the fishing effort and of the abundance estimates.
5) You mentioned abundance estimates of the porpoises, but I don’t think you used this information.
6) You mentioned stranding increased, especially in years with low fisherman delivered bycaught animals. You could consider adding the numbers of stranding to the observed bycatch. However, some of the stranded animals did die of other causes.
7) You said you wanted to help managers but most of these variables would be difficult to develop a management action.
8) Lunar phase appears to be important, so it would be very informative if you dove more into why. But perhaps if you used bycatch rate then this would not be important if more fishing is occurring at this time.

Validity of the findings

1. All of the bycatch underlying data were provided. Although the fishing effort data were not

2. Stated conclusions follow from the analysis, but as discussed above, the analysis is very limited and I bet there is more in the data that should be shared.

Reviewer 2 ·

Basic reporting

Please see "General comments for the author" section below for my review.

Experimental design

Please see "General comments for the author" section below for my review.

Validity of the findings

Please see "General comments for the author" section below for my review.

Additional comments

Review of “What influences harbor porpoise bycatch in the
south-western Baltic Sea?”

Summary
This paper is an interesting summary of ~30 years of harbor porpoise bycatch data in Baltic Sea gill nets. There are several important findings here that are work reporting in the primary literature. In my estimation, those key points are: 1) 90% of all bycaught animals were four years or younger 2) 69.3% of all porpoises were bycaught from July to November 3) The majority of bycatch (28.6%) occurred during the lunar phase 5. Also, possibly (but not definitely) that this bycatch has declined since 1996. However, there are some major issues I have with the manuscript and therefore cannot endorse its publication at present. Please address those comments and I would be happy to reconsider the paper.


Major points:

• I tend to think paper titles are best when presenting the main finding. Consider changing title to “Age of individual and lunar cycle predict harbor porpoise bycatch in the south-western Baltic Sea” or something along those lines.
• For the meteorological data analyzed were you able to get information on local cloud cover or just wind-speed. Cloud cover would be an important factor to consider as it could negate the luminosity provided by a full (or nearly full) moon.
• For the vessels considered (boats >8m), were fishermen required to report bycatch, and/or were there independent fisheries’ observers on-board? If neither is true how can you be sure that bycatch was reported honestly and accurately? At minimum this should be specified in the methods section.
• Statistical analyses are only briefly described and therefore challenging to determine whether the approach taken was appropriate. My initial thought is that a simple ANOVA is overly simplistic for these data, and that a mixed model approach would be more suitable. Please justify your current statistics better (e.g., cite/describe other bycatch studies that used similar statistics) or use a different statistical approach with better justification.
• Lines 167-168: “The majority of bycatch (28.6%) occurred during the lunar phase 5”. This is an interesting and useful finding, please conduct and report statistics on this finding.
• Lines 176-178: I agree this is a key finding, but your vague discussion of “Within this critical time period, mitigation measures should be applied” is unhelpful. What critical time period? The animals are juveniles for ~4 years with overlapping generations. Do they (the juveniles and adults) spatially segregate? Are there any specific ideas you can point to that would reduce bycatch risk to this vulnerable life-stage?
• Lines 182-193: this is an important issue (voluntary fisherman reporting) that calls into question a potential key finding (bycatch numbers declining post 1996) of your work. Is there any data from other regions (ideally NE Atlantic) that suggests that marine mammal/gill-net bycatch declined from pre-mid-1990s to post-mid-1990s? I understand that you cannot say for sure whether this change has to do with a difference in reporting by fisherman, which is unfortunate, but here would be helpful to draw on other studies from the same region that are better controlled for reporting.
• Figure 4B I recommend a log-scale (or some form of data transformation/visualization technique) where the reader will be able to see the majority of the data more easily.
• One of the most interesting points of the paper is your claim that a statistically significant portion of bycatch events happened during bright moonlight. In addition to not tracking cloud cover (which may have been impossible, but seems important to make this claim), in Figure 5 your data show that yes, the most bycatch happened during full moon conditions; however, the second most bycatch events happened during new moon conditions without moonlight. How do you explain that?



Minor points:
Passive voice should be changed to active voice when possible. Several examples:
• Lines 90-91: “a long-term data-set… was analyzed”
• Line 93: “it was examined”
• Lines 112-113: “data from dedicated aerial surveys… were used”
• Lines 295-296: “how bycatch… is affected by”
Imprecise language at points that could be made more professional. Several examples:
• Lines 44-45: “gill nets, which is the main cause of death in marine mammals,” may be true for pinnipeds and odontocetes, but not for mysticetes.
• Line 67: describing bycatch rates as “way above”. How much greater?
• Line 256: “way lower” How much lower? Be specific.
Lines 207-208: “During necropsies at ITAW over the last 30 years, only single animals showed scars indicating a previous bycatch situation.” This sentence doesn’t make sense as is, please reword.
Line 288-289: “of two captive showed” I believe the word “porpoises” is missing?

---

## Round 0.2 · Major Revisions

We have received two reviews for your study. Unfortunately, we were not able to secure reviews from both original reviewers. The additional review received from an expert in the field found substantial merit in the study, but also noted important points that deserve further revisions. While I am conscious of the risk of creating a moving target during repeated review stages, I strongly feel that the new issues raised and the need for clarifications emphasized by reviewer #3 are important and, if properly dealt with, will result in a greatly improved manuscript.

Reviewer 2 ·

Basic reporting

The authors have addressed my concerns and I consider this paper acceptable for publication

Experimental design

The authors have addressed my concerns and I consider this paper acceptable for publication

Validity of the findings

The authors have addressed my concerns and I consider this paper acceptable for publication

Additional comments

The authors have addressed my concerns and I consider this paper acceptable for publication

Reviewer 3 ·

Basic reporting

See general comments to the authors.

Experimental design

See general comments to the authors.

Validity of the findings

See general comments to the authors.

Additional comments

Review of “Age and lunar cycle predict harbour porpoise bycatch in the south-western Baltic Sea.

Overall: This manuscript makes use of a long-term data set on bycatch in the south-western Baltic Sea to attempt to gain more information on the impact of environmental and anthropogenic factors on the chances of bycatch. This information is important and much needed. Overall, the manuscript is well written. Unfortunately, there are a number of areas where a lot of clarification is needed before this manuscript is ready for publication. My main concerns are highlighted below followed by more specific comments on the sections of the manuscript.

Main concerns:
- The Baltic region has three distinct populations of harbour porpoises, with very different conservation statuses, that all exhibit distinct seasonal movement patterns. I think the interpretation of the results could benefit from a clearer explanation of how porpoise presence in this region is influenced by the movement patterns of the two populations likely to utilise the area observed, and how this may influence the seasonal pattern in bycatch rate reported. i.e. Belt Sea population mainly uses the area in summer, yet Baltic Proper population is thought to move into the area over the winter months. Making it clear how these results likely relate to which porpoise populations is important from a management perspective.
- Linked to this, there needs to be consistency in when the author is referring to a specific population, or all porpoises in the Baltic region. For example, in the abstract “a decline in Baltic porpoises”- is this in relation to the Baltic Proper population, or all porpoises in the Baltic region. This should be checked and clarified throughout. As far as I understand, none of the bycaught animals have been tested genetically to assign population ID, so it is not possible for the authors to say for sure which population the bycaught animals belong to- but this should at least be discussed as a factor to consider.
- In regards to lunar influences, I wonder why the lunar cycle at the locations was not calculated (e.g. using the simple “lunar” package in R), rather than relying on values from a single station? Also, in order to reduce the number of factors/levels in the analysis (which is always a good thing), I wonder whether it would have been clearer to lump phases of the lunar cycle with similar illumination level (e.g. waxing and waning) together? For example, in Figure 6 with the histogram of lunar phase and bycatch, it would have shown a clear trend across five bars of lunar illumination level. Also, studies showing the impact of light on bycatch should probably be discussed. Here is an example for birds (Field et al. 2019- High contrast panels and lights do not reduce bird bycatch in Baltic Sea gillnet fisheries). There is also a lot of literature showing light reduces fish bycatch. I am unsure what exists for harbour porpoises, but it should be discussed given the findings surrounding higher bycatch during the full moon (higher light levels).
- Exclusion of years with no bycatch (2005, 2010, 2014) as it was unlikely that none occurred in those years makes it seem as though a different assumption was applied to these years relative to the other 27. In other years you don’t account for bycatch that was likely not reported either…
- Fishing effort data is only for a short time period (3 years). This detail should be made clearer, as it is not a 30 year data set for this analysis.
- The analysis section of the methods needs a lot more detail added. As I understand, you plan to compare days with and without bycatch, yet this would likely result in a lot of zeros (thousands of days over 30 years with no bycatch). However, in figures 4 and 5 I see that you only compared to 60 or so days with no bycatch? How were these selected? And why were the number of days with bycatch different between the wind and precipitation analysis?
- Chisquared tests were used, but no definition of the expected outcome was given, or justification for why this was the assumed expected outcome. This information is essential.
I also have some more specific comments, including:

Abstract:
Line 46: There was “a” significant….
Line 49: During “a” full moon. Check throughout. Should be “a full moon” or “the full moon” depending on sentence context.

Introduction:
Line 64-67: Long convoluted sentence: suggest changing it to “For all three populations in the Baltic region, assessed…” Could also be built into more detail on the three populations and their movements as highlighted in concerns above.
Line 70-72: Also use of alternative gear is another option- (see Königson et al 2015. Cod pots in the Baltic. Are they efficient and what affects their efficiency? )
Line 84-87: Lunar influence on cetaceans has been seen also. E.g. The echolocation of indo-pacific humpback dolphins (Wang et al. 2015- Passive Acoustic Monitoring the Diel, Lunar, Seasonal and Tidal Patterns in the Biosonar Activity of the Indo-Pacific Humpback Dolphins (Sousa chinensis) in the Pearl River Estuary, China) and pilot whales changed the dive behaviour and spatial area use (Owen et al. 2019- Lunar cycles influence the diving behavior and habitat use of short-finned pilot whales around the main Hawaiian Islands). This should be mentioned here, not just that it influences fish, and may influence porpoises.
Line 92: Should be made clear here that you mean harbour porpoise abundance.

Methods:
Line 104-106: Should be split into two sentences for clarity. Option 1 was age using teeth. Option 2 was based on length. Right now it reads as though the length option was a second step of option 1…which I don’t think is right. Please reword.
Line 111: delete the comma.
Line 115-117: This sentence is not clear and needs a reword. What was it monthly averages of? Fishing effort?
Line 120: change “whether” to “if”- flows better than “whether weather”.
Line 123: What was the temporal scale that precipitation data that were used ? Daily mm? Same for cloud cover. Was it daily average for the day of the bycatch event?
Line 127-130: Why was a single location used for lunar data? It is possible to obtain information on lunar cycles for exact locations and times. E.g. using the “lunar” package in R.
Line 131-134: In analyses the levels of factors can influence results, with it typically being better to minimise levels as much as possible. I wonder if it would have been clearer to lump lunar phases with similar illumination levels (e.g. waxing and waning gibbous), or if there is a reason why you did not do this?
Line 138: Make it clear that the abundance assessment is not over the full 30 year period.
Line 142: It is not clear that multiple models were run, and that model selection was needed? Also, it is not common to use statistical significance as an assessment of model fit. I think the explanation of which models, and model structures were actually tested, as this section is not clear.
Line 143: “days with and without bycatch”. You only had 140 bycatch events over 30 years, so this implies that you had 10,810 days without bycatch. This would result in a massive zero inflation problem and a very unbalanced sample size, and it is not clear how you got around this problem. I see in figure 4 and figure 5 that a much smaller sample size (n around 60) is given, but it is unclear how you reached this number and which days were selected. Also why less days are considered than were available for days with bycatch?
Line 144: Chi squared tests require a comparison between an expected and observed outcome. It is essential that you define what your expected outcomes were in these tests, and what you based these assumptions on.

Results:
Line 154-155: delete the sentence starting with “in 2005, 2010...” as this information is already given above. See comment above on the validity of this.
Line 155-156: This increase to 8 per year seems higher than the mean of 7.4 in the first decade. Potential reasons for this increase in recent years should be discussed.
Line 161: Is this 73% of the 90% mentioned above? These % and differences are a little unclear. Consider re-wording for clarity.
Line 164: “no significant difference”- give the statistic results.
Line 177-178: This result goes in line with my suggestion above to lump all lunar phases with similar illumination levels.

Discussion:
Line 189-191: See comment above on the increase in bycatch to 8 animals a year after 2015 and the need to discuss this.
Line 194: What mitigation measures in EU waters are you referring to? This should be specified here for the reader. Was there something specific in the mid 90s (other than a reduction in bycatch reporting) that could have reduced fishing effort/bycatch? Also consider indirect measures, e.g. cod ban in recent years that could have influenced fishing effort.
Line 195: catch limits of what?
Line 206: change word “demand” to “imply”. This sentence seems quite speculative and should be made clear that this is the case. You could similarly speculate that they have seen other animals bycaught and understand the danger.
219: Could be good to include information on when young animals likely separate from their mother, and whether juvenile animals move around alone, or in mixed age groups.
Line 223-225: So does this mean that juveniles potentially aren’t at higher risk than adults, it is just that the age distribution of the animals bycaught matches the age distribution of the animals out there.
Line 236: add ‘s’ to the end of ‘occur’. Make it clear that it is the monthly distribution in this specific area. As it is likely different in other areas depending on fishing effort/porpoise movement patterns.
Line 239: Again- make it clearer this is only in your area. There are plenty of areas where porpoise density is highest in winter and overlapping with areas used for gillnet fishing.
Line 245-249: I think the fact you only used 3 years of fishing effort data should be addressed here. It is possible that the distribution of fishing effort changed over the 30 years, and that a larger overlap of fishing effort with the time that porpoises are present in the area leads to higher bycatch.
Line 256: delete the word prey.
Line 260: noise from factors such as shipping has been shown to reduce the echolocation of porpoises and cause fleeing behaviour (Dyndo 2015, Wisniewska et al. 2018- already in the reference list)- reduced echolocation could lead to less nets detected.
Line 275: why would they be more active during a full moon? I don’t follow. Porpoises need to forage continuously so it is not as though they can stop foraging outside of a full moon period? Active in what way?
Line 276: you ruled out an impact of weather on bycatch, so it is unlikely that weather influenced this. Or is this why you found an influence? i.e. nets only set in low wind and no rain- so the lack of influence is due to only sampling one end of the range of possible weather? Was there an interaction between weather and fishing effort?
Line 277: You have fishing effort data and lunar cycle information. Did you see this interaction in the data?
Line 278: give more specifics of what you mean by “clear lunar activity patterns”
Line 283: Captive porpoises are not subjected to the possible influence of lunar cycles on their prey, so it makes sense that they would not display lunar differences.
Line 295: The sentence is not clear and needs a reword.
Line 298: delete the ‘s’ off the end of ‘results’.

Figure 1: Change “bycatches” to “bycatch events”- change throughout the manuscript as “bycatches” is a weird word in English, and it is better to be consistent throughout the document (e.g. with figure 4 and 5).

Figure 2 and Table 1: that some bycatch events were excluded from the analysis should be made clear in the methods/results section, not only in the figure.

Figure 3: If I interpret this correctly, the highest fishing effort was in 37G1, where there was hardly any bycatch. This should also be discussed, particularly in light of the distribution of the 2 populations in this area. What was the fishing effort in 38G1, 37F9 and 38F9?

Figure 4: how were the number of days with no bycatch determined? Why were these 52 or 68 days selected out of the 10,800 ish available with no bycatch, and why were there less days with bycatch than 140 (i.e. precipitation n =115 days with bycatch, but wind speed n = 75 days with bycatch)? This should be explained in the methods/results too.

Figure 5: Seems to be a weird repeat of part of figure 4B?

---

## Round 0.3 · Minor Revisions

The authors have done a good job in further revising their manuscript. There are now only a few minor issues regarding statistics (potential zero inflation problem and additional clarifications of sample size and statistics being used) and management recommendations that were further raised by the reviewer and that need to be integrated in the next version.

Reviewer 3 ·

Basic reporting

Line 67-71- I feel it is worth noting that for the Critically Endangered Baltic Proper population it has been estimated that a maximum bycatch of 0.7 individuals is able to be sustained- meaning that bycatch of this population should be at zero.

Experimental design

The authors have done an excellent job of clarifying the statistics. I have one remaining question, and that is whether zero inflation can be an issue with chi squared tests, or the regression analyses? The data set used were heavily zero inflated, and while I am not sure if this is an issue with the tests used, it should be clarified/stated in the text what (if any) issue it is likely to create, and how it was dealt with.

Figure 4: it is still unclear why 75 bycatch events were analysed for wind speed, but 115 for precipitation. Yet the methods say only "Five animals were excluded from the spatial analysis due to unknown bycatch location". And the methods also say there was a total of 140 bycaught animals. I am also a little confused about how you tested "for significant differences between wind speed and precipitation during days with and without bycatch, we used ANOVA Chi-squared tests"? What was your expected wind speed against measured wind speed? I assume these data would have been better suited to the polynomial models mentioned above? Is this a mistake in writing?

Validity of the findings

I also have concerns on the management recommendations in the conclusion. "To reduce bycatch, fishing efforts could be reduced seasonally between e.g., July and November, especially in areas with high bycatch of juveniles and neonates, such as Eckernförde Bight and Schlei". Based purely on bycatch data I can see why this would be recommended. But as I mentioned previously, I think the authors need to be careful with what is recommended here, as the area may be used by the Critically Endangered population of harbour porpoises during the winter months (we don't currently know how far west they go in the winter, and there is no genetic data on which animals were bycaught (great that this is now highlighted!). Even one bycaught harbour porpoise from the Baltic Proper population is a massive issue for this population. I think this issue should be mentioned in the discussion, and again in the conclusion. (i.e. that the potential for the critically endangered population to use the area in the winter months should be taken into account).

Additional comments

The authors have done a good job of incorporating most of my concerns/comments and I congratulate them on pulling it together.

---

## Round 0.4 · accepted · Accept

I am happy with the final revisions made to the manuscript.